# Treatment of High-Strength Animal Industrial Wastewater Using Photo-Assisted Fenton Oxidation Coupled to Photocatalytic Technology

**Jae Hong Park \*, Dong Seok Shin and Jae Kwan Lee**

Water Environmental Research Department, National Institute of Environmental Research (NIER), Gyeongseo-dong, Seo-gu, Incheon 22689, Korea

\* Correspondence: jhong@korea.ac.kr

**Abstract:** Animal wastewater is one of the wastewaters that has a color and is difficult to treat because it contains a large amount of non-degradable organic materials. The photo-assisted Fenton oxidation technique was applied to treat animal wastewater, and the optimal conditions of chemical oxygen demands (COD) removal were analyzed according to changes in pH, ferrous ion, $H_2O_2$, and ultraviolet (UV) light intensity as a single experimental condition. Experimental results showed that, under the single-factor experimental conditions, the optimal conditions for degradation of animal wastewater were pH 3.5, Fe(II) 0.01 M, $H_2O_2$ 0.1 M, light intensity 3.524 mW/m$^2$. Under the optimal conditions, COD removal efficiency was 91%, sludge production was 2.5 mL from 100 mL of solution, color removal efficiency was 80%, and coliform removal efficiency was 99.5%.

**Keywords:** photocatalytic and photo-assisted Fenton oxidation; animal wastewater; advanced oxidation process (AOPs)

---

## 1. Introduction

Animal wastewater generated in animal breeding facilities such as for cattle and pigs contains a high concentration of non-degradable organic matter, nutrients (nitrogen and phosphorus), and color, which can have a negative effect on the water environment when discharged without being properly processed in the treatment facilities. The amount of animal wastewater is not higher than that of domestic sewage and industrial wastewater, but the load of pollutants is 50–150 times higher than that of municipal wastewater in Korea [1].

Animal wastewater is being treated by various methods, such as conventional biological treatment. However, since animal wastewater contains a high concentration of non-degradable organic matter, the existing treatment methods have difficulty in treating animal wastewater and have not achieved a satisfactory treatment effect.

Animal wastewater has been treated alone or combined with municipal wastewater and night soil in Korea [2]. However, their treatment capacities are insufficient as compared with animal wastewater generation. Therefore, it is thought that the excess is discharged to soil, farmland and watershed as it is untreated [2]. After all, inappropriate disposal of animal wastewater may pose environmental problems such as the accumulation of pollutants in soil and pollution of ground and surface water due to leaching and run-off of pollutants [2]. For this reason, livestock wastewater is considered a main water pollutant source recently in Korea.

However, the problem of livestock wastewater is not only a problem in Korea but also a global problem. According to a survey, 60% of river pollution and 45% of lake pollution are caused by livestock wastewater [3].

Advanced oxidation processes (AOPs) are a commonly used method for the treatment of inert matter, toxic mater, pesticides, etc. in wastewater. Above all, the advantage of AOP is that the byproducts after the reaction produce harmless materials that do not require secondary environmental pollution or treatment [4].

Water treatment methods using AOP include photocatalysts, ozone, Fenton reagents, photo Fenton reagents, etc. The AOP-applied water treatment technique is to produce OH radicals, which are powerful oxidation species in the chemical reaction process, and to oxidize pollutants in wastewater into harmless materials.

Many studies on the degradation of non-biodegradable, toxic and hazardous substances were carried out using the Fenton and Photo-Fenton reactions [5–9]. However, few studies have been carried out on the treatment of real animal wastewater using these methods.

The purpose of this study was to evaluate the applicability of photo-assisted Fenton method for the treatment of real animal wastewater. The effect of pH, $H_2O_2$, ferrous salt and ultraviolet (UV) light on the degradation of organic pollutants, color, and fecal coliform were investigated. In addition, the comparison of the existing Fenton method with chemical oxygen demands (COD), color and coliform removal efficiency was also performed.

## 2. Materials and Methods

### 2.1. Chemical Reagents

The photocatalyst used in this study was commercial $TiO_2$ (Degussa P-25, 5 g/L), in power form. P25 has a primary particle size of 21 nm, a surface area of $50 \pm 15$ m$^2$/g and its crystalline mode is 99.5% anatase. The pH was controlled with an accuracy of $\pm 0.02$ by adding either sodium hydroxide (reagent grade, Duksan) or sulfuric acid (reagent grade, Duksan). Ferrous sulphate heptahydrate ($FeSO_4 \cdot 7H_2O$, 99.5%) from Duksan, hydrogen peroxide ($H_2O_2$, 30 wt%) from Merck, and ferric chloride ($FeCl_3$, 97%) from Samchon were used.

### 2.2. Photocatalytic and Photo-Assisted Fenton Reactor

The schematic diagram of the photo-assisted Fenton experimental set-up used in this study was shown in Figure 1. It consists of an irradiation source and photoreactor. All the experiments were carried out in a continuous flow through cylindrical quartz columns (each diameter 10 mm and length 700 mm) with recirculation of the solution. The light source is furnished with 40 W UV lamp (Sankyo Denki Co., Ltd., Kanagawa, Japan, 1200 mm length, 32 mm diameter), mounted on standard fluorescent tube holders. The column was exposed to a luminous source composed of UV lamps with a maximum emission at 254 nm. The UV intensity was controlled by turning on different number of UV lights and the maximum intensity was 5.742 mW/cm$^2$. The effective volume of photoreactor (550 mm (width) $\times$ 640 mm (length) $\times$ 110 mm (height)) was 1.88 L. The photoreactor bottom was wrapped in aluminum to reflect any illumination to the column.

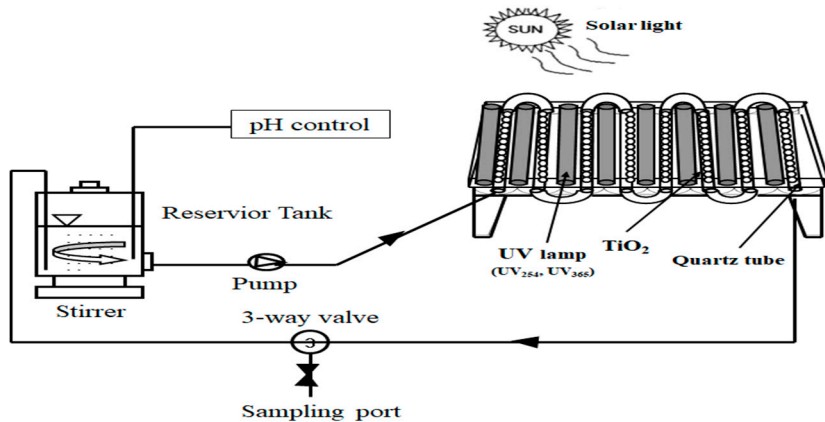

**Figure 1.** Schematic diagram of the photo-assisted Fenton reactor.

### 2.3. Wastewater Preparation and Experimental Procedures

The animal wastewater was obtained from an animal wastewater treatment plant located in Kyonggi province in Korea. The animal wastewater pretreated by coagulation with 3000 mg/L $FeCl_3$ to remove suspended solid (SS) components because it can interfere with UV light transmission before using it. $FeCl_3$ was injected into animal wastewater, and the supernatant was used for experiment after 10 min slow stirring at 100 rpm and 30 min of precipitation. The characteristics of the pretreated wastewater were as follows: chemical oxygen demand (COD) 1450 mg/L, color 0.1853 abs and coliform $1.2 \times 10^{10}$ number/100 mL.

For the photo-assisted Fenton experiment, the animal wastewater (8 L) was contained in a 10 L glass container and was stirred. The wastewater circulated photo-assisted Fenton reactor (Figure 1) through modules at a flow rate of 1.5 L/min. The modules are connected in series and the wastewater flows directly from one to the other and finally to the reservoir container. At the end of each equilibrium time, the $TiO_2$ was removed from suspension by centrifugation (Combi-514R, HANIL Science Industrial, Gimpo, Korea) at 3500 rpm for 10 min. For the analysis, a 50 mL aliquot was taken at various intervals, and then the COD, color and fecal coliform were measured.

Control COD removal experiments were conducted in 8 L animal wastewater, spiked with initial COD of 1450 mg/L. In these experiments, COD removals by individual UV, $TiO_2$, $UV/TiO_2$, $Fe/H_2O_2$, $UV/Fe/H_2O_2$, $TiO_2/Fe/H_2O_2$ and $UV/TiO_2/Fe/H_2O_2$ systems were tested and compared to understand the mechanism of treatment process.

### 2.4. Analytical Methods

UV intensity was measured primarily with a radiometer (VLX-3W Radiometer 9811-50, Cole Parmer lnstrument Co., Saint Neots, UK) at 254 nm. However, it is impossible to directly measure the UV intensity reaching the surface of the animal wastewater because the reactor used in this experiment has wastewater flowing through the installed cylindrical quartz columns. Therefore, to measure the UV intensity reaching the surface of animal wastewater, a flat plate quartz having the same material and thickness as the cylindrical quartz columns used in the experiment was used. The UV intensity measured directly below the plate quartz can be considered to have the same value as the UV intensity irradiated on the animal wastewater flowing in the cylindrical quartz columns. The average UV intensity through animal wastewater in this study was measured to be 3.524 $mW/cm^2$.

pH was determined using a pH meter (Orion 525A). Color was determined using a UV–Vis spectrometer (UV-1201, Shimadzu Co., Kyoto, Japan). Animal wastewater decolorization was expressed as a decrease in absorbance of the wavelength at maximum absorbance of the inlet wastewater.

The potassium dichromate closed reflux colorimetric method was used to measure the COD [10]. The fecal coliform was measured using the standard method [10].

The COD removal (%) achieved was calculated according to Equation (1):

$$(COD_{inlet} - (COD_{outlet}/COD_{inlet})) \times 100 \qquad (1)$$

where $C_{inlet}$ and $C_{outlet}$ are COD concentration values of animal wastewater at initial and a given time t, respectively.

Color removal was determined based on the maximum absorbance of UV–Visible spectrum using Equation (2):

$$\text{Animal wastewater decolorization} = 1 - (C_t/C_0) \times 100\% \qquad (2)$$

Coliform was measured with the membrane filtration method using C-EC agar, and the results were as coliform forming units (CFU) in 100 mL.

## 3. Results and Discussions

### 3.1. COD Removal by Different Degradation Systems

Figure 2 depicts the removal efficiencies of COD by different degradation systems, i.e., UV, $TiO_2$, $UV/TiO_2$, $Fe/H_2O_2$, $UV/Fe/H_2O_2$, $TiO_2/Fe/H_2O_2$ and $UV/TiO_2/Fe/H_2O_2$. Results revealed COD was rarely removed when only $TiO_2$ and UV were used. In the photocatalytic reactions combined with $TiO_2$ and UV, the electrons emitted by the reaction reacted with oxygen in water to generate OH ($^{\bullet}OH$) and superoxide ($O_2^{\bullet-}$) radicals, resulting in the decomposition of organic matter in animal wastewater, which increased COD removal efficiency (Equations (3)–(7)). $TiO_2$ is a photoactive material that upon illumination with a suitable light source (with energy ≥ band gap ~3.2 eV), electron ($e^-$)-hole ($h^+$) pairs are generated at the interfacial surface (Equation (3)) [11,12]. In this process, the valence-band (VB) electrons ($e^-_{VB}$) are excited by the incident light and migrate to a higher energy level (i.e., conduction-band) ($e^-_{CB}$), leaving behind positively-charged holes ($h^+_{VB}$) [11]. The surface $e^-_{CB}$ was reported to interact with dissolved oxygen in aqueous media and generate reactive oxygen species (e.g, $O_2^{\bullet-}$, $HOO^{\bullet}$ and $^{\bullet}OH$) (Equation (7)) [11].

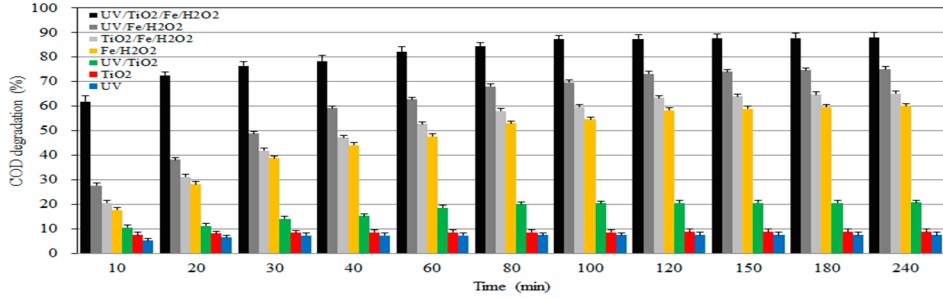

**Figure 2.** The degradation of chemical oxygen demands (COD) under different treatment processes.

In Fenton oxidation ($Fe/H_2O_2$), the COD removal efficiency was increased by decomposing organic matter by OH radicals produced by reaction of hydrogen peroxide and divalent iron ions (Equation (9)). The COD removal efficiency by the Fenton oxidation reaction was higher than that of $TiO_2$ photocatalytic reaction ($TiO_2/UV$) because the UV could not pass through the wastewater sufficiently because the livestock wastewater was high. The COD removal efficiency by the Fenton oxidation reaction was higher than $TiO_2$ photocatalytic reaction ($TiO_2/UV$) because the livestock wastewater has high organic concentration and color, so UV could not pass through the wastewater sufficiently and it is difficult to react with $TiO_2$. In the case of injecting $TiO_2$ photocatalyst into the Fenton oxidation process, it showed a higher COD removal efficiency than the Fenton oxidation reaction, which is judged to be due to the organic adsorption effect of $TiO_2$. When UV was irradiated to the Fenton oxidation process, COD removal efficiency was improved. The COD removal rate was highest in this process because OH radicals were generated in both the Fenton oxidation and $TiO_2$ photocatalytic reactions.

The highest COD removal rate was observed in the $UV/TiO_2/Fe/H_2O_2$ process, which is a combination of Fenton oxidation and $TiO_2$ photocatalytic reaction (Equation (10)). In the $UV/TiO_2/Fe/H_2O_2$ process, COD removal efficiency was nearly to 90% at 100 min of reaction time.

$TiO_2$ photocatalytic reaction can be expressed as [13]:

$$TiO_2 + hv \rightarrow TiO_2(e^-_{CB} + h^+_{VB}) \tag{3}$$

$$TiO_2(e^-_{CB} + h^+_{VB}) \rightarrow TiO_2(recombination) \tag{4}$$

$$TiO_2(h^+_{VB}) + OH^-_{sur} \rightarrow {}^\bullet OH \tag{5}$$

$$TiO_2(e^-_{CB}) + O_2 \rightarrow TiO_2 + O_2{}^{\bullet-} \tag{6}$$

$$e^-_{CB} + O_2 \rightarrow O_2{}^{\bullet-} \rightarrow HOO^\bullet \tag{7}$$

$${}^\bullet OH \text{ (or } O_2{}^{\bullet-}) + \text{Organic pollutant} \rightarrow CO_2, H_2O, NO_3{}^-, SO_4{}^{2-}, \text{etc.} \tag{8}$$

Fenton oxidation can be expressed as [14]:

$$Fe^{2+} + H_2O_2 \rightarrow Fe^{3+} + OH^- + {}^\bullet OH \tag{9}$$

Photocatalytic and photo-assisted Fenton oxidation can be expressed as:

$$TiO_2 + hv + Fe^{2+} + H_2O_2 \rightarrow TiO_2 + Fe^{3+} + OH^- + {}^\bullet OH + O_2{}^{\bullet-} \tag{10}$$

### 3.2. The Effect of pH

The pH value has an important effect on the oxidation potential of OH radicals because of the reciprocal relation of the oxidation potential to the pH value ($E^o$ (pH = 0) = 2.8 V (at 25 °C) and $E^{14}$ (pH = 14) = 1.95 V) [6,15–18].

In general, the Fenton reaction is known to be effective between pH 3 and 5 because iron ions are in divalent state ($Fe^{2+}$) under acidic conditions, so this study also conducted experiments under acidic conditions of pH less than 7, i.e., pH 2.5, 3.5, 5.5, 6.5. In the study of Barrera-Salgado et al. (2016) [19], the oxidation potential pH 5.0 and pH 7.0 of OH radicals were lower than that at pH 2 and pH 3. They said that at pH > 3, the formation of hydroxyl radicals slows down because of hydrolysis of Fe(III) and $Fe^{3+}$ precipitation as $Fe(OH)_3$ from the solution.

Under basic conditions above pH 7, COD removal efficiency is reduced. Because the leaching iron ions can form iron hydroxide sludge $Fe(OH)_3$, which eventually reduces the production of OH radicals. In addition, $H_2O_2$ becomes unstable as pH increases, which has a negative effect on the production of OH radicals.

As depicted in Figure 3, COD removal efficiency was relatively high in pH 3.5–4.5 range and low removal efficiency in pH 5.5–6.5 range. In the range of pH 5.5 or more, $Fe^{2+}$ was converted to $Fe^{3+}$ or other form of Fe, which decreased rapidly and consequently, the Fenton oxidation reaction was limited, which led to low COD removal efficiency.

Additionally, with the pH as low as 2.5, the degradation effect decreased. This is because a low pH reduces the stability of $H_2O_2$ and may release a proton to form hydronium ($H_3O^+$), enhancing its stability; however, this can reduce the reactive activity of $Fe^{2+}$ when it participates in the degradation process [20].

The optimum pH condition in this study was 3.5, and it showed 87.9% COD removal efficiency and took about 100 min to reach equilibrium.

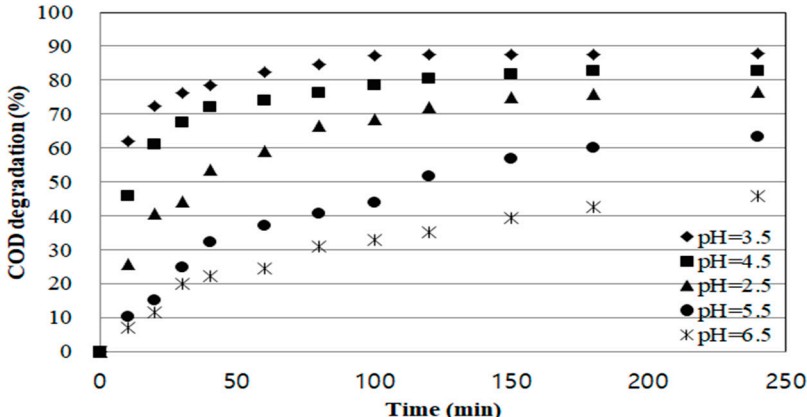

**Figure 3.** The influence of pH on the degradation of COD ($Fe^{2+}$ = 0.01 M, $H_2O_2$ = 0.1 M, reaction time = 240 min, $UV_{254}$ intensity = 3.524 mW/cm$^2$).

### 3.3. The Effect of Ferrous Ion

If Fe(II) does not exist sufficiently in the Fenton reaction, pollutants cannot be effectively decomposed. However, if Fe(II) is too much, it has a negative effect on the Fenton reaction. This is because when Fe(II) is injected in excess, sludge generation after the reaction increases and OH radical regeneration occurs. Also, a higher addition of Fe(II) resulted in a brown turbidity that hinders the absorption of the UV light required for photolysis and causes the recombination of OH radicals. In this case, Fe(II) reacts with OH radicals as a scavenger [21].

$$^{\bullet}OH + Fe^{2+} \rightarrow OH^- + Fe^{3+} \tag{11}$$

In this study, the amount of Fe(II) injected was tested in the range of 0.0025 to 0.05 M. As shown in Figure 4, the removal efficiency of COD increased as the amount of Fe(II) increased. The optimal COD removal efficiency was 89.7% at Fe(II) 0.01 M and reached equilibrium condition after 90 min.

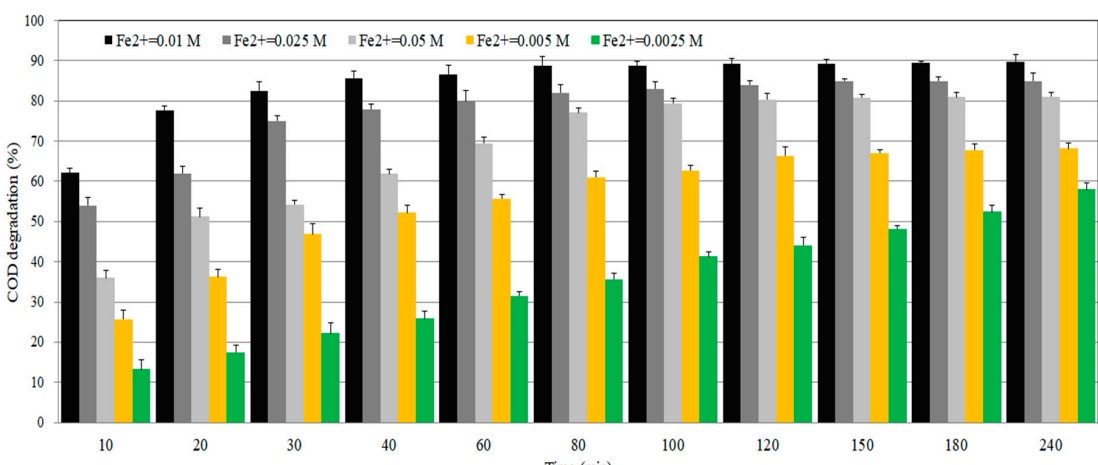

**Figure 4.** Effect of $Fe^{2+}$ dosage on COD removal in photo-assisted Fenton oxidation (pH = 3.5, $H_2O_2$ = 0.1 M, $UV_{254}$ intensity = 3.524 mW/cm$^2$).

However, when the amount of Fe(II) were increased to 0.025 M and 0.05 M, COD removal efficiencies were as 85.2% and 81.1%, which were lower than that of 0.01 M. Therefore, COD removal efficiency can be lowered even if Fe(II) is excessively injected, and excessive Fe(II) injection can increase unnecessary costs. It is desirable that the ratio of Fe(II) should be as small as possible, so that the recombination can be avoided and the sludge production from the iron complex is also reduced.

### 3.4. The Effect of $H_2O_2$

If the proper amount of $H_2O_2$ exists, COD removal efficiency can be increased. In the presence of electron acceptor, hydrogen peroxide produces OH radicals, such as the reaction of Equation (12) [2]. Additionally, when hydrogen peroxide absorbs ultraviolet light of less than 300 nm, OH radicals are produced such as Equation (13) [22].

$$H_2O_2 + e^- \rightarrow {}^\bullet OH + OH^- \tag{12}$$

$$H_2O_2 + hv \rightarrow 2{}^\bullet OH \tag{13}$$

OH radicals produced by reactions of Equations (11) and (12) photodegrade organic matter as strong oxidants. Therefore, the COD removal efficiency in animal wastewater is determined by how much OH radical is produced by photo-assisted Fenton reaction.

However, if hydrogen peroxide is injected too much, COD removal efficiency may not be high because of the autodecomposition of $H_2O_2$ and the recombination of OH radicals (${}^\bullet OH$) as indicated in Equation (14) [23].

$$ {}^\bullet OH + H_2O_2 \rightarrow H_2O + HO_2{}^\bullet \tag{14}$$

In this experiment, the concentration range of hydrogen peroxide is 0.1 M from 0.01 M. As shown in the results of Figure 5, the removal efficiency of COD increased as the concentration of hydrogen peroxide increased. The optimum result was obtained when the concentration of hydrogen peroxide was 0.1 M, and the COD removal efficiency rate was about 91% at 60 min of reaction time.

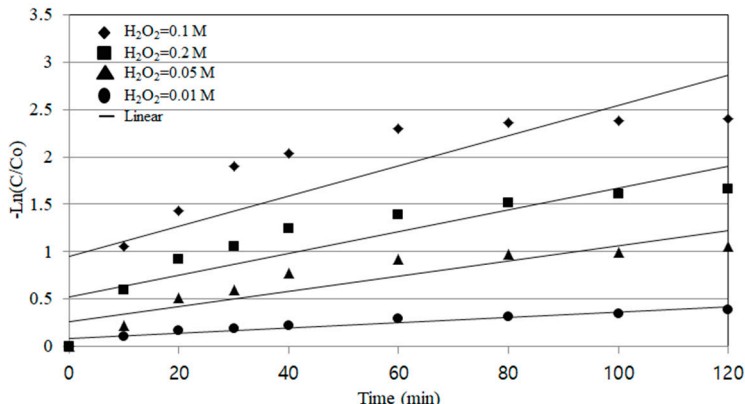

**Figure 5.** Effect of $H_2O_2$ dosage on COD removal in photo-assisted Fenton oxidation (pH = 3.5, $Fe^{2+}$ = 0.01 M, $UV_{254}$ intensity = 3.524 mW/cm$^2$).

However, the COD removal efficiency was lower in 0.2 M than in 0.1 M. This may be because the hydrogen peroxide was excessively injected and the $H_2O$ and $HO_2{}^\bullet$ were formed due to the recombination of OH radicals.

### 3.5. The Effect of UV Light Intensities

If UV are transmitted to the Fenton oxidation reaction, it reacts with the hydrogen peroxide as seen in Equation (14), producing OH radicals, so pollutant removal is much more advantageous than the Fenton oxidation reaction.

In addition, UV is more effective in COD removal because it reacts with $TiO_2$ photocatalyst used in this experiment as seen in Equations (3)–(8) [22,24] and produces strong oxidation species of superoxide and OH radical. The first step of the photocatalytic reaction is the generation of electrons and holes in the $TiO_2$ particles irradiated with UV according to the reaction formula presented in Equation (3). In the next step, the recombination of $TiO_2$ particles occurs with the emission of heat as in Equation (4). However, when dissolved oxygen or electron donors are present, the OH radicals

are formed by reactions between the valence band holes ($h^+$) and the active OH group or $H_2O$ on the surface, as shown in Equation (5). In the next step, super-oxide ($O_2^{\bullet-}$) is produced by the reaction of the conduction band electron ($e^-$) generated by the photochemical reaction and dissolved oxygen, such as Equation (6), is generated. Hydroxyl radicals ($^\bullet OH$) and superoxide ($O_2^{\bullet-}$) is a strong reactive species related to photocatalytic decomposition of many organic compounds, which results in the degradation of organic compounds (Equation (8)).

In order to understand the COD removal characteristics according to UV intensity, the experiment was performed under the optimal conditions of the previous experiment. In other words, the experiment results were shown in the Figure 5 by adjusting the UV intensity to 0.467–5.742 mW/cm$^2$ under the reaction conditions of pH 3.5, ferrous ion 0.01 M, and $H_2O_2$ 0.1 M.

As shown in Figure 6, COD removal efficiency is higher as the intensity of UV increases because the generation of oxidized species is active due to the reaction between ultraviolet light and hydrogen peroxide, ultraviolet light and $TiO_2$. The average COD removal efficiency is 36.7%, 62.9%, 89.5% and 92.9%, respectively, as the intensity of ultraviolet light becomes stronger to 0.467–5.742 mW/m$^2$.

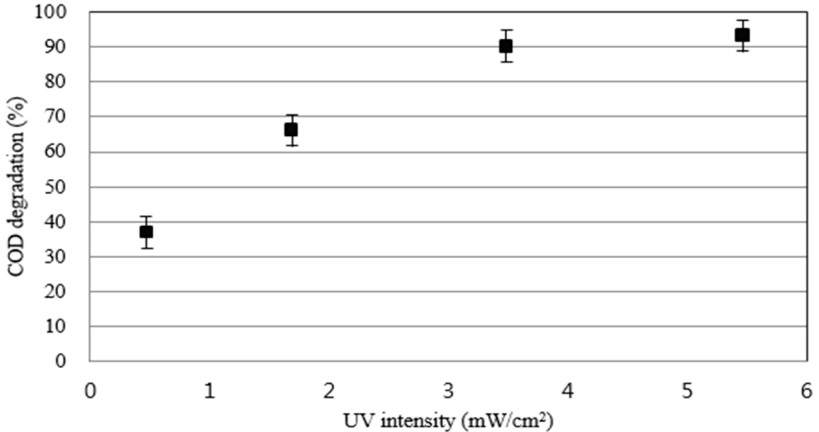

**Figure 6.** Effect of UV light intensities on COD removal in photo-assisted Fenton oxidation (pH = 3.5, $Fe^{2+}$ = 0.01 M, $H_2O_2$ = 0.1 M).

Although the highest COD removal efficiency was shown at 5.742 mW/m$^2$, which has the largest UV intensity, the proper UV intensity is 3.524 mW/m$^2$, considering the economical part such as electricity costs.

### 3.6. The Effect of Light Types

To understand the COD removal characteristics according to the light type, the types of light sources were tested under five conditions: $UV_{254}$ (1.70 mW/cm$^2$), $UV_{365}$ (1.68 mW/cm$^2$), and solar (clear sky, partly cloudy sky, and thick cloudy sky). The solar light intensity was measured using a radiometer with a UV-A (365 nm). All the experiments were performed under random weather conditions, between 10:00 AM and 4:00 PM, in Korea (38° N latitude). Figure 7 shows that the solar light intensity on a sunny day slowly increased from 0.64 mW/cm$^2$, at 10:00 AM, to 1.72 mW/cm$^2$ at noon, which decreased after 2:00 PM.

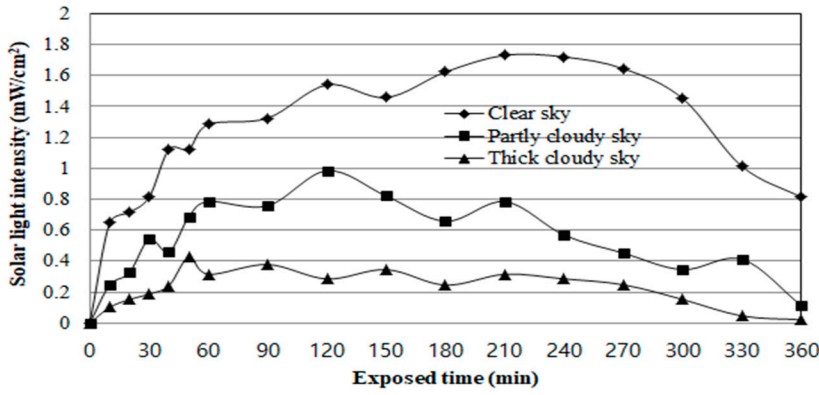

**Figure 7.** Solar light intensity vs. exposed time.

COD removal efficiency of animal wastewater was investigated by adjusting the UV intensity of $UV_{254}$ lamp and $UV_{365}$ lamp at a level similar to the maximum UV intensity at sunny sky. As shown in Figure 8, COD removal efficiency was the highest when $UV_{254}$ lamp was used. The second highest COD removal efficiency was observed at the clear sky. In the case of using $UV_{365}$ lamp, the COD removal rate was the third highest, followed by partly cloudy sky and thick cloudy sky.

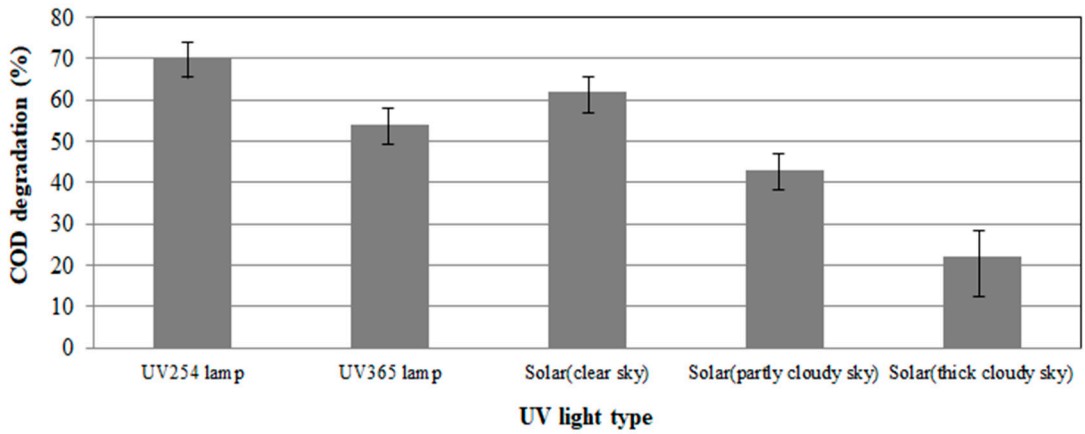

**Figure 8.** Effect of UV light types on COD removal in photo-assisted Fenton oxidation (pH = 3.5, $Fe^{2+}$ = 0.01 M, $H_2O_2$ = 0.1 M).

In case of clear sky, overall UV intensity was lower than that of $UV_{365}$ lamp, but COD removal efficiency was higher. When solar light is used as an ultraviolet light source, it is considered that it emits various kinds of ultraviolet rays in addition to $UV_{365}$. It is presumed that these ultraviolet rays are involved in the reaction and generate OH radicals and the like to increase the COD removal efficiency.

### 3.7. Sludge Production

Ferrous ion is a necessary chemical for the formation of OH radical in the Fenton oxidation process, but if ferrous ion is excessively present, it can be said that it is not necessarily advantageous for the removal of pollutants by escaping the appropriate $Fe/H_2O_2$ ratio, and unnecessary chemical costs are required. Therefore, when the pollutants are to be treated by using the Fenton oxidation reaction, it is necessary to investigate the appropriate amount of ferrous ion injection so that secondary environmental problems caused by ferrous ion do not occur and high cost is not required.

In this study, the amount of ferrous ion injection was changed to 0–0.05 M to investigate the amount of sludge generation. Other factors were tested under the optimal conditions suggested in the previous experiment.

After the experiment, 500 mL of sample was collected in 500 mL glass cylinder and precipitated for 30 min, and the volume of sludge in cylinder was measured by volume ratio.

As shown in the results of Figure 9, sludge generation tends to increase continuously as the amount of ferrous ion injection increases. The amount of ferrous ion injection was relatively small until 0.1 M, but when ferrous ion is injected more than 0.02 M, the sludge generation rapidly increases. Therefore, it is appropriate to maintain the injection amount of ferrous ion less than 0.1 M under the same experimental conditions as this study.

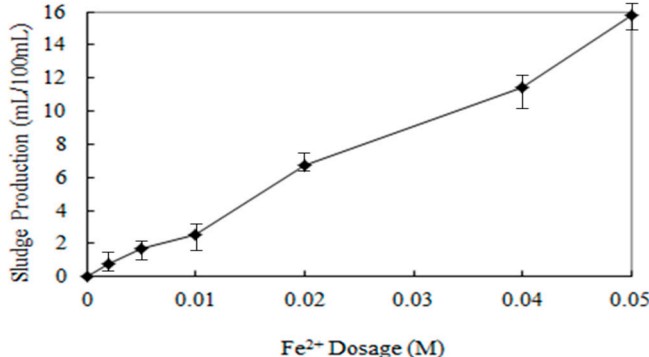

**Figure 9.** Production of sludge with $Fe^{2+}$ dosage (pH = 3.5, reaction time = 100 min, $H_2O_2$ = 0.1 M, $UV_{254}$ intensity = 3.524 mW/cm$^2$, settling time = 30 min).

### 3.8. The Effect of Fe(II) on the Color Removal Efficiency

Animal wastewater has brown color, so it is very likely that complaints will occur if the color is not sufficiently removed at the treatment facility and discharged to the river.

The color in animal wastewater can be oxidized and removed by OH radicals in the Fenton oxidation reaction. However, because iron salt is used in the Fenton oxidation reaction, excessive injection of iron salt may cause color. Therefore, it is necessary to investigate the appropriate amount of iron salt for color removal in the Fenton oxidation reaction. In addition, injection of too much iron salt can have a negative impact on over-spending drug costs.

In order to determine the effects of Fe(II) on the color removal efficiency, experiments were carried out at various Fe(II) addition (0–0.05 M).

As shown in Figure 10, the color removal efficiency was increased with increasing Fe(II) dosage. Even at no addition of Fe(II), the color removal was 16%. This reduction may be due to a photolysis of $H_2O_2$ when hydrogen peroxide absorbs the UV light with a wavelength < 300 nm (Equation (3)) and the direct photolysis of the organic pollutants. By the direct photolysis of the organic pollutants the molecules may reach an excited state and can be oxidized partly by the oxygen.

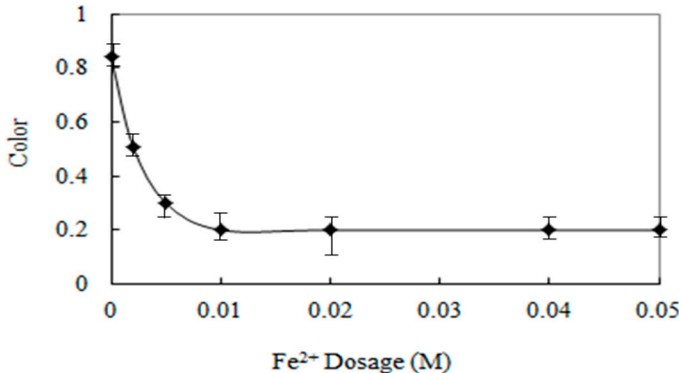

**Figure 10.** Variation of color removal with $Fe^{2+}$ dosage in photo-assisted Fenton oxidation (pH = 3.5, $H_2O_2$ = 0.1 M, $UV_{254}$ intensity = 3.524 mW/cm$^2$).

However, the color removal efficiency was not influenced by Fe(II) dosage above 0.02 M. This can be explained that a higher addition of Fe(II) resulted in a brown turbidity that hinders the absorption of the UV light. Thus, 0.01 M was selected as the optimal $Fe^{2+}$ concentration.

### 3.9. The Coliform Removal Efficiency

Animal wastewater discharge standards include microbial items, which are set differently for each country. Some define only coliforms (e.g., Japan), and others define both coliforms and *E. Coli* (e.g., the United States). In Korea, the effluent standard for coliform is set up at the animal manure wastewater treatment facility. Therefore, this study was also aimed to investigate the degree of coliform in animal manure wastewater by Fenton oxidation reaction. The coliform removal efficiency was investigated under optimal conditions. As shown in the results of the experiment, coliform removal efficiency increased rapidly at the beginning of UV irradiation time (Figure 11). The removal efficiency was 92% at 5 min of UV irradiation time and 99% or more at 25 min. After 40 min, coliform was almost completely removed. Therefore, the proper reaction time for coliform removal can be set appropriately according to the effluent standard between 10 and 40 min.

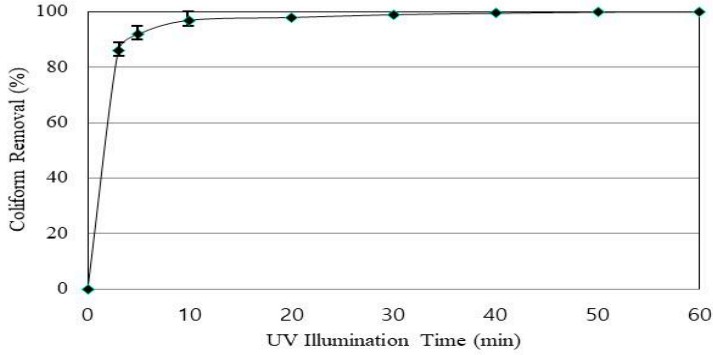

**Figure 11.** Variation of coliform removal with UV illumination time in photo-assisted Fenton oxidation at the optimal conditions (pH = 3.5, $H_2O_2$ = 0.1 M, $UV_{254}$ intensity = 3.524 mW/cm$^2$).

### 3.10. Comparison of Fenton and Photocatalytic and Photo-Assisted Fenton

To determine the effect of photo-assisted Fenton, a comparative experiment with the Fenton oxidation method was conducted (Figure 12). The results of the experiment under the optimal conditions (reaction time = 100 min, pH = 3.5, $H_2O_2$ = 0.1 M, $Fe^{2+}$ dosage 0.01 M, $UV_{254}$ intensity = 3.524 mW/cm$^2$, settling time = 30 min) showed that the removal efficiency of pollutants by photo-assisted Fenton was higher than that of Fenton. When the animal wastewater was treated by the photo-assisted Fenton method, the removal efficiency of COD, color, and coliform was about 35%, 20%, and 5% higher than that of the Fenton method. The difference in removal efficiency of coliform was not large, because coliform can be easily removed by oxidation species (OH radical) that occurs in the reaction process as well as UV light.

On the other hand, in the AOP oxidation process using $H_2O_2$, it is necessary to investigate the residual $H_2O_2$. This is because, when $H_2O_2$ is injected more than necessary, COD may be increased due to the remaining $H_2O_2$ which is not decomposed. Also, if the subsequent treatment process is a biological treatment method, the $H_2O_2$ remaining in the microorganism may have a negative effect. This residual hydrogen peroxide may be harmful to humans.

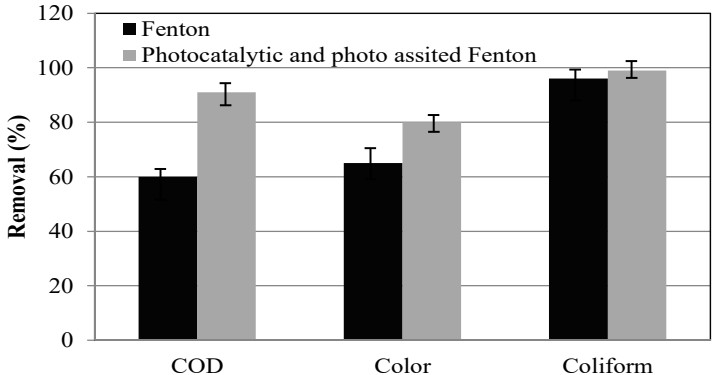

**Figure 12.** Comparison of COD, color, and coliform removal efficiency between Fenton and photo-assisted Fenton under optimal condition.

The concentration of residual hydrogen peroxide was analyzed with a fluorescence detector using the fluoromatric method. The fluorometer was operated at 310 nm and 430 nm for extinction and emission wave lengths, respectively. Analysis of residual $H_2O_2$ concentration showed no residual $H_2O_2$ after the reaction and did not affect the measured values such as COD and coliform.

In the optimal operation condition experiment of Fenton and photocatalytic and photo-assisted Fenton, the residual hydrogen peroxide remaining in animal wastewater was not detected. Therefore, it does not affect the analysis value such as the COD value by the residual $H_2O_2$.

## 4. Conclusions

The experimental study showed that the photo-assisted Fenton oxidation can effectively degrade COD, color and coliform in animal wastewater. Therefore, it is necessary to establish appropriate reaction conditions to treat this type of wastewater—that is, a high concentration of organic matter. The presence of UV light in the Fenton oxidation reaction was capable of enhancing the oxidation reaction.

The advantage of the photo-assisted Fenton process as an oxidative pre- or post-treatment step over other photochemical oxidation processes is efficiency, especially if not much suspended solids are present. In general, the treatment of animal wastewater is being conducted by using conventional biological treatment but effective removal of COD, color and coliform was not achieved. For the practical application based on our results, we suggest that combination of photo-assisted Fenton process with biological treatment is a promising alternative. It is considered that if the suspended solids (SS) which can act negatively (UV transmission interference, etc.) in the photo-assisted Fenton process are removed by the primary treatment with the conventional biological treatment method, and the organic matter and color which are difficult to remove from the biological treatment are removed by applying the photo-assisted Fenton process technique secondarily, it will be used as an effective animal wastewater treatment method.

**Author Contributions:** Conceptualization, J.H.P.; methodology, J.H.P.; data curation, J.H.P.; writing—original draft preparation, J.H.P.; writing—review and editing, J.H.P.; supervision, D.S.S., J.K.L.

**Funding:** This research was supported by the National Institute of Environmental Research under Grant Nos. 1900-1946-303.

**Acknowledgments:** This research was conducted as a regular research activity in National Institute of Environment Research (NIER-RP2013-271).

**Conflicts of Interest:** The authors declare no conflicts of interest.

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
