# Peer review of "Treatment of High-Strength Animal Industrial Wastewater Using Photo-Assisted Fenton Oxidation Coupled to Photocatalytic Technology"

_water, doi:10.3390/w11081553_

Round 1
Reviewer 1 Report
The authors should explain better the wastewater preparation, since these kind of wastewater should contain high amount biological organic matter and high amount of solids.
Why the authors do not follow the BOD5? This wastewater contain a huge amount of biodegradable compounds.
The authors said " Above all, the advantage of AOP is that the by products after the reaction produce harmless materials that do not require secondary environmental pollution or treatment [4]." How the authors can prove this issue? you cannot generalize this sentence.
Figures will be improved. The markers are very similar.
Why the authors do not follow the H2O2 residual concentration?
Why the authors have not tested other radiations?
With 30 minutes of reaction can the authors ensure that all the iron is precipitated?
The authors cannot also follow the E.Coli?
Author Response
Response to Reviewer 1 Comments
Point 1 :Why the authors do not follow the BOD5? This wastewater contain a huge amount of biodegradable compounds.
Response 1:
As you pointed out, there are a lot of high concentrations of organic matter, BOD5, in the animal wastewater. However, the reason why COD rather than BOD5 is analyzed in this paper is that non-biodegradable or inert organic substance is also present in large quantities in animal wastewater.
Animal wastewater is mainly treated in biological treatment method. Non-biodegradable organic matter can not be sufficiently removed by biological treatment method, which may cause secondary environmental problems when animal wastewater is treated at wastewater treatment plant and then discharged to agricultural land (used as fertilizer) or river.
And AOP (Fenton, Photo assisted Fenton, UV/TiO2, etc.) is a method used to treat non-biodegradable materials that are difficult to treat biologically, so COD was selected as an experimental item in this study.
Point 2 :The authors said " Above all, the advantage of AOP is that the by products after the reaction produce harmless materials that do not require secondary environmental pollution or treatment [4]." How the authors can prove this issue? you cannot generalize this sentence.
Response 2:
This sentence is a phrase commonly cited in many other papers. In other words, as shown in Eq. 7, when non-biodegradable and toxic organic substances are treated with AOP, CO2, NO3, H2O, etc. are generated as by-products, which means that when they are discharged into the environment, they do not need additional treatment and are not environmentally pollutants.
Point 3 :Figures will be improved. The markers are very similar.
Response 3:
We changed the display method of some pictures in consideration of your comments.
Point 4 :Why the authors do not follow the H2O2 residual concentration?
Response 4:
In consideration of your review, we added an explanation of the remaining H2O2 in the paper.
Point 5 :Why the authors have not tested other radiations?
Response 5:
We added the results of other radiations to reflect the comments.
Point 6 :With 30 minutes of reaction can the authors ensure that all the iron is precipitated?
Response 6:
Figure 9 does not mean that all iron is precipitated in 30 minutes. The measurements of Figure 9 were the result of maintaining the same condition for other parameters and increasing the amount of iron injection to 0, 0.002, 0.005, 0.01, 0.02, 0.04, 0.05 M and then precipitating for 30 minutes.
If the precipitation time is increased to more than 30 minutes, sludge may be generated more, but in this paper, the precipitation time is set to 30 minutes and compared.
Figure 9 is to show that if the amount of iron injected in the photo assisted Fenton experiment is increased to more than the appropriate amount, the amount of sludge generated after the reaction can be increased.
Point 7 :The authors cannot also follow the E.Coli?
Response 7:
E. coli was not taken into consideration in this paper because the criterion for effluent discharge is fecal coliform in animal wastewater treatment facilities In Korea,
Reviewer 2 Report
The abbreviation COD is first time mentioned in abstract and then is used in text (for example line 57) but you explain this term first time on the line 83. Please, explain this abbreviation in text where it was mentioned for the first time.
In Abstract and also in the text you mentioned "removal rate" in %. The unit "%" is more proper for the term "removal efficiency", for removal rate you must use different unit. Please, correct it in the manuscript.
Line 83 and line 90: In English are used the decimal point for numbers. Please, check it out also in the rest of the article.
Line 99: The oxidation potential for different pH should be listed more clearly. Values of pH as an upper index are confusing. Moreover, the literary source of this information cannot be listed as an upper index.
Figure 2: The highest oxidation potential is for the most acid environment (E(pH~0)=2.8V). Why is the more acid environment, with pH=2.5, worse than pH=3.5 and pH=4.5?
Figs. 2-4 and 6: Why you connected the experimental points with line? The lines are without any additional information. In Fig 5 you have a data without a line it look more clearly.
Figs. 2-5: Please, edit the figures. Y-axis has incorrect scale (no decimal number) and the legends in the figures also must have stated upper and lower index on an appropriate expression.
Line 164 and 201: In the text you have a pH of experiments on the value 3.5 but in the figures description you have pH=5. Please, correct the inaccuracy.
Chapter 3.2 The effect of ferrous ion: Why was the two highest Fe2+ doses with such a big step? You doubled each Fe2+ dosage for first three experiments and then you used 5x higher dose. Why? Can you added the measurement for the dosage Fe2+ = 0.02M or 0.025M?
Line 210, 232, 126: If you mentioned more numbers but only one unit you must have the numbers in the bracket. Without brackets, the unit is valid only for the last number. Please, correct it in the whole manuscript.
Chapter 3.8: What were the conditions of the experiments that compared each other? Please, added this information to the manuscript.
Author Response
Response to Reviewer 2 Comments
Point 1: The abbreviation COD is first time mentioned in abstract and then is used in text (for example line 57) but you explain this term first time on the line 83. Please, explain this abbreviation in text where it was mentioned for the first time.
Response 1:
We explained the abbreviation to the first place in consideration of your comments.
Point 2: In Abstract and also in the text you mentioned "removal rate" in %. The unit "%" is more proper for the term "removal efficiency", for removal rate you must use different unit. Please, correct it in the manuscript.
Response 2:
In consideration of your comments, we modified the "removal rate" in % to "removal efficiency".
Point 3: Line 83 and line 90: In English are used the decimal point for numbers. Please, check it out also in the rest of the article.
Response 3:
We revised the text in consideration of your comments.
Point 4: Line 99: The oxidation potential for different pH should be listed more clearly. Values of pH as an upper index are confusing. Moreover, the literary source of this information cannot be listed as an upper index.
Response 4:
We revised the sentence considering the contents of the comment and added some references.
Unfortunately, we could not find any literature on oxidation potentials for different pH.
All the literatures that deal with the oxidation potential of OH radicals show the value in the range (Eo = 2.8 V and E14 = 1.95 V) or only the maximum value (Eo = 2.8 V).
Literature Cases in Range:
1) Bauer, R.; Waldner, G.; Fallmann, H.; Hanger, S.; Klare, M.; Krutzler, T. The photo-fenton reaction and the TiO2/UV process for wastewater treatment –novel developments. Catalysis Today, 1999, 53, 131-144. https://doi.org/10.1016/s0920-5861(99)00108-x
2) Deng, Y.; Zhao, R. Advanced Oxidation Processes (AOPs) in Wastewater Treatment, Water Pollution 2015, 1(3), 167–176. DOI 10.1007/s40726-015-0015-z
3) Badawy, M.I.; Ghaly M.Y.; Gad-Allah, T.A. Advanced Oxidation Processes for the Removal of Organophosphorus Pesticides from Wastewater, Desalination 2006, 194 (1-3), 166–75. https://doi.org/10.1016/j.desal.2005.09.027
A case of literature represented by maximum value:
1) Grote, B. APPLICATION OF ADVANCED OXIDATION PROCESSES (AOP) IN WATER TREATMENT, 37th Annual Qld Water Industry Operations Workshop, 2012, Page No. 21. file:///C:/Users/user%20pjh/AppData/Local/Microsoft/Windows/INetCache/IE/2WXNHQEV/Bill_Grote.pdf
2) Wei Li, W.; Zhou, Q,; Hua, T. Removal of Organic Matter from Landfill Leachate by Advanced Oxidation Processes: A Review, International Journal of Chemical Engineering 2010, Volume 2010,10 pages. http://dx.doi.org/10.1155/2010/270532
Point 5: Figure 2: The highest oxidation potential is for the most acid environment (E(pH~0)=2.8V). Why is the more acid environment, with pH=2.5, worse than pH=3.5 and pH=4.5?
Response 5:
Most of the literature shows that the Fenton reaction is optimal when the pH is about 3; at this pH, the solution can produce a large number of Fe(OH)+ ions, which has an activity higher than Fe2+.
(Malik, P.K.; Saha, S.K. Oxidation of Direct Dyes with Hydrogen Peroxide Using Ferrous Ion as Catalyst. Sep. Purif. Technol. 2003, 31, 241–250.)
However, with the pH as low as 2.5, the degradation effect decreased. This is because a low pH reduces the stability of H2O2 and may release a proton to form hydronium(H3O+), enhancing its stability ;however, this can reduce the reactive activity of Fe2+ when it participates in the degradation process.
(Inmaculada, V.G.; Jesús, J.L.P.; Manuel, S.P.; José, R.U. Comparative study of oxidative degradation of sodium diatrizoate in aqueous solution by H2O2/Fe2+, H2O2/Fe3+, Fe (VI) and UV, H2O2/UV, K2S2O8/UV. Chem. Eng. J. 2014, 241, 504–512.)
We added an explanation for your inquiry in the text.
Point 6: Figs. 2-4 and 6: Why you connected the experimental points with line? The lines are without any additional information. In Fig 5 you have a data without a line it look more clearly.
Response 6:
We revised the Figures in consideration of you and other reviewer comments.
Point 7: Figs. 2-5: Please, edit the figures. Y-axis has incorrect scale (no decimal number) and the legends in the figures also must have stated upper and lower index on an appropriate expression.
Response 7:
We revised the Figures in consideration of you and other reviewer comments.
Point 8: Line 164 and 201: In the text you have a pH of experiments on the value 3.5 but in the figures description you have pH=5. Please, correct the inaccuracy.
Response 8:
It’s our errata. It was revised.
Point 9: Chapter 3.2 The effect of ferrous ion: Why was the two highest Fe2+ doses with such a big step? You doubled each Fe2+ dosage for first three experiments and then you used 5x higher dose. Why? Can you added the measurement for the dosage Fe2+ = 0.02M or 0.025M?
Response 9:
In consideration of your comments, we modified the Figure 4 by adding 0.025 M of Fe2+ injection.
Point 10: Line 210, 232, 126: If you mentioned more numbers but only one unit you must have the numbers in the bracket. Without brackets, the unit is valid only for the last number. Please, correct it in the whole manuscript.
Response 10:
We revised the text in consideration of your comments.
Point 11: Chapter 3.8: What were the conditions of the experiments that compared each other? Please, added this information to the manuscript.
Response 11:
We added experimental conditions in the text as your comments.
Reviewer 3 Report
Major revision. The topic is interesting but there is lack of details on experimental conditions in several parts as well as poor discussion and low quality of graphics. The English language of the manuscript need intensive revision as it is not up to standards. Here are some comments about the work:
- It looks like the employed process is hybrid between photocatalyic (UV/TiO2) and homogenous photo-Fenton reaction. I suggest reconsider the referral to the treatment process to accomodate both processes.
- Please indicate in-details how the wastewater pretreatement with FeCL3 was performed. How did the COD was measured? Were did they obtained the wastewater samples? Did they run control experiments in clean water to compare the efficiency of the current system with those conducted in wastewater matrix?
- It is unclear whether the researchers checked their results reproducibility? Where are the error bars/standard error for their results?
- It is a flow reactor, then, a radiometer won't be an accurate tool to measure the actual light intensity reaching the solution surface. A calibration of the UV intensity by actinometry or ferric oxalate method is needed in this case, you may see these publications for more details: https://doi.org/10.1016/j.jhazmat.2014.07.041; https://doi.org/10.1016/j.cej.2015.12.046.
- Equation 1 is incorrect, please revise and correct it.
- The individual contribution of UV, TiO2, UV/TiO2, Fe/H2O2, UV/Fe/H2O2, TiO2/Fe/H2O2 and UV/TiO2/Fe/H2O2 systems for the removal of COD are needed to understand the mechanism of treatment process.
Author Response
Response to Reviewer 3 Comments
Point 1: It looks like the employed process is hybrid between photocatalyic (UV/TiO2) and homogenous photo-Fenton reaction. I suggest reconsider the referral to the treatment process to accomodate both processes.
Response 1:
In consideration of your comments, we added a description of the two processes in section 3.1 Blank Experiment.
We have also revised the title in consideration of this.
Point 2: Please indicate in-details how the wastewater pretreatement with FeCL3 was performed. How did the COD was measured? Were did they obtained the wastewater samples? Did they run control experiments in clean water to compare the efficiency of the current system with those conducted in wastewater matrix?
Response 2:
We supplemented the paper by adding the comments you gave.
The experimental results of this paper are measured by considering the result of blank tests (control experiments) using the third distilled water.
Point 3: It is unclear whether the researchers checked their results reproducibility? Where are the error bars/standard error for their results?
Response 3:
In consideration of your comments, we modified the picture by adding an error bar to the experimental results.
Point 4: It is a flow reactor, then, a radiometer won't be an accurate tool to measure the actual light intensity reaching the solution surface. A calibration of the UV intensity by actinometry or ferric oxalate method is needed in this case, you may see these publications for more details:https://doi.org/10.1016/j.jhazmat.2014.07.041; https://doi.org/10.1016/j.cej.2015.12.046.
Response 4:
We calibrated UV intensity in consideration of your comments.
Point 5: Equation 1 is incorrect, please revise and correct it.
Response 5:
It’s our errata. It was revised.
Point 6: The individual contribution of UV, TiO2, UV/TiO2, Fe/H2O2, UV/Fe/H2O2, TiO2/Fe/H2O2 and UV/TiO2/Fe/H2O2 systems for the removal of COD are needed to understand the mechanism of treatment process.
Response 6:
We added the comments you made to the text.
Round 2
Reviewer 2 Report
For the Figures 2, 3, 5, 7: The Legend should be fill with a white opaque colour. The lines passing through the legend deteriorate the readability.
Reviewer 3 Report
The manuscript can be considered for publication in Waters after accommodating the following comments:
There are still several typos and grammatical errors in the papers that need revision: for example,
- Page 1, line 16: the capitalization of “Light Intensities” and using plural form “s” while referring to only one value of light intensity needs modification
- P1, line 16-17: There should be a space between “numbers” and “scientific unit”, please follow up with the remaining text
Page 3, line 96: the light intensity reaching the surface of water was 3.54 mW/cm2, it is more practical and realistic to use this value in the abstract in leu of the one measured by the radiometer. The following papers can be cited as a reference for the calibration method https://doi.org/10.1016/j.jhazmat.2014.07.041, https://doi.org/10.1016/j.cej.2015.12.046, https://doi.org/10.1016/j.jhazmat.2018.06.062
Page 3, line 103: change “Where C0 and Ct are concentration values of animal wastewater” into “Where C0 and Ct are COD concentration values of animal wastewater”
Page 3, line 105-110: You don’t need to make a new section for control experiments. Move this part to be one paragraph at the end of “section 2.3” as follows:
(Control COD removal experiments were conducted in 8 L animal wastewater, spiked with initial COD of 1450 mg/L. In these experiments, COD removals by individual UV, TiO2, UV/TiO2, Fe/H2O2, UV/Fe/H2O2, TiO2/Fe/H2O2 and UV/TiO2/Fe/H2O2 systems were tested and compared to understand the mechanism of treatment process.)
Page 3, line 113-118:
- Change the section title from “Blank experiment” into “COD removal by different degradation systems”
- Change “The experiments were conducted in the presence of UV, TiO2, UV/TiO2, Fe/H2O2, UV/Fe/H2O2, TiO2/Fe/H2O2 and UV/TiO2/Fe/H2O2 (Figure 2)” into “Figure 2 depicts the removal efficiencies of COD by different degradation systems, i.e., UV, TiO2, UV/TiO2, Fe/H2O2, UV/Fe/H2O2, TiO2/Fe/H2O2 and UV/TiO2/Fe/H2O2”
- Change “The experiment results of the Fig. 2 showed that COD was…..” into “Results revealed COD was………”
- The radical sign should be always at oxygen atoms, this is the correct form “•OH” and “O2•―”, please correct accordingly in the whole text.
Page 3, line 118: The following relevant article can be cited to refer to the chemical interaction at TiO2 surface and formation of different radicals “ https://doi.org/10.1016/j.apcatb.2018.09.039”
Page 5, Equation 10: correct the charge of iron ion on the right-hand of equation to be “3+”
Page 5, line 172-177: Please revise the language here. The way the sentences were written is totally no technical and non-professional. Do not start your sentences with (or even use) the phrase “If……….”. Please consider passing the manuscript to A PROFESSIONAL ENGLISH SPEAKER!
Page 5, line 185: What is meant with “…..and unnecessarily large amount of medicines would be injected”; it is totally unclear?
Page 7: For experiments involved changing light intensity, since the radiometer reads a totally different UV intensity than the actual one reaching the solution (as you indicated by actinometry and ferrioxalate method), then how you did measure these intensities?
Page 9: For color removal experiments, how was the color being measured? In other words, what was the analytical technique.
Page 10, line 307: I suggest lessens the specific reference to “Korea” in your references, it definitely limits the audience for the research you done. Instead, try to find something more internationally “such as international guidelines from different countries or from EPA and WHO to refer to. This way may increase the benefit of your research!
Page 10: How were the Coliform experiment was performed and what was the analytical technique used for measurements?
Page 11, Line 329-330: This is totally non-technical. The primary purpose of your experiment is to achieve maximum removal of COD. This may require a certain dose of H2O2 regardless of its effect on your analytical method for total COD measurement in the system. I suggest if you want to effectively lessens the impact of residual H2O2 that you measure the TOC removal instead!
Page 11, line 333: Complete details of H2O2 experimental conditions, measurements and results are needed in the main text.
Round 3
Reviewer 3 Report
The manuscript can be published in Water.